# Coordination of LMO7 with FAK Signaling Sustains Epithelial Integrity in Renal Epithelia Exposed to Osmotic Pressure

**DOI:** 10.3390/cells11233805

**Published:** 2022-11-28

**Authors:** Yen-Yi Zhen, Chien-Hsing Wu, Hung-Chun Chen, Eddy Essen Chang, Jia-Jung Lee, Wei-Yu Chen, Jer-Ming Chang, Pei-Yun Tseng, Yue-Fang Wang, Chi-Chih Hung

**Affiliations:** 1Division of Nephrology, Department of Internal medicine, Kaohsiung Medical University Hospital, Kaohsiung Medical University, Kaohsiung 80708, Taiwan; 2Division of Nephrology, Department of Internal Medicine, Kaohsiung Chang-Gung Memorial Hospital, Kaohsiung 83301, Taiwan; 3College of Medicine, Chang-Gung University, Taoyuan 33303, Taiwan; 4School of Medicine, College of Medicine, Kaohsiung Medical University, Kaohsiung 80708, Taiwan; 5Regenerative Medicine and Cell Therapy Research Center, Kaohsiung Medical University, Kaohsiung 80708, Taiwan; 6Institute for Translational Research in Biomedicine, Kaohsiung Chang Gung Memorial Hospital, Kaohsiung 83301, Taiwan; 7Department of Biochemistry and Molecular Biology, College of Medicine, National Cheng Kung University, Tainan 83701, Taiwan; 8Department of Medical Research, Kaohsiung Medical University Hospital, Kaohsiung Medical University, Kaohsiung 80708, Taiwan; 9School of Post-Baccalaureate Medicine, College of Medicine, Kaohsiung Medical University, Kaohsiung 80708, Taiwan

**Keywords:** LMO7, secretome, FAK hypertonicity, osmotic stress, epithelial integrity, epithelial barrier, cortical stress fiber

## Abstract

The kidney epithelial barrier has multifaceted functions in body fluids, electrolyte homeostasis, and urine production. The renal epithelial barrier (REB) frequently faces and challenges osmotic dynamics, which gives rise to osmotic pressure (a physical force). Osmotic pressure overloading can crack epithelial integrity and damage the REB. The endurance of REB to osmotic pressure forces remains obscure. LMO7 (LIM domain only 7) is a protein associated with the cell–cell junctional complex and cortical F-actin. Its upregulation was observed in cells cultured under hypertonic conditions. LMO7 is predominantly distributed in renal tubule epithelial cells. Hypertonic stimulation leads to LMO7 and F-actin assembly in the cortical stress fibers of renal epithelial cells. Hypertonic-isotonic alternation, as a pressure force pushing the plasma membrane inward/outward, was set as osmotic disturbance and was applied to test FAK signaling and LMO7 functioning in maintaining junctional integrity. LMO7 depletion in cells resulted in junctional integrity loss in the epithelial sheet-cultured hypertonic medium or hypertonic-isotonic alternation. Conversely, FAK inhibition by PF-573228 led to failure in robust cortical F-actin assembly and LMO7 association with cortical F-actin in epithelial cells responding to hypertonic stress. Epithelial integrity against osmotic stress and LMO7 and FAK signaling are involved in assembling robust cortical F-actin and maintaining junctional integrity. LMO7 elaborately manages FAK activation in renal epithelial cells, which was demonstrated excessive FAK activation present in LMO7 depleted NRK-52E cells and epithelial integrity loss when cells with LMO7 depletion were exposed to a hypertonic environment. Our data suggests that LMO7 regulates FAK activation and is responsible for maintaining REB under osmotic disturbance.

## 1. Introduction

In metazoans, cells frequently state in an erratic extracellular environment with dynamic osmosis [1,2]. When cells endure osmotic pressure changes, osmotic adaptation serves as a physiological mechanism to adapt to and withstand osmotic challenges [3,4,5]. As osmotic adaptation does not work well, cells shrink/swell and cell shape changes occur [6,7]. Confronting extracellular hypertonicity or hypotonicity, cytoskeletal remodeling is one of the osmotic adaptation mechanisms to withstand osmotic pressure force, maintain cell shape, and preserve tissue integrity [3,8].

The osmolarity in the intrinsic environment of the kidneys is variable [2]. Changes in osmolarity in the kidney elicit an osmotic pressure force that pushes the plasma membrane inward or outward [6,8,9,10]. Tubulointerstitial milieu appear regularly with a steeper osmotic gradient along the medulla, from a relatively lower osmotic milieus in the corticomedullary junction to a high osmotic environment in the papilla tip [1,11]. This osmotic gradient benefits urine concentration and water and electrolytes reabsorption into the blood vessels [12]. And osmotic gradient derived dynamics of osmolarity is challenging renal cells all the time. The epithelial cells plasma membrane in the renal tubule was frequently forced inward or outward by osmotic pressure force when environmental osmolarity represented dynamic change. There is inference of renal cells possessing specialized osmotic tolerance from the violent dynamic osmolarity in kidney. Responding to osmotic pressure force, cytoskeletal remodeling in renal epithelial cells is a vital mechanism for withstanding osmotic pressure force and maintaining cell morphology and epithelial integrity [3]. Mounting evidence has demonstrated that focal adhesion kinase (FAK) is a central mediator driven by physical forces, such as osmotic pressure, to preserve epithelial integrity. [13,14,15].

FAK activation represents a physiological cascade and pathological programs in the kidney [16,17,18,19]. FAK activation inhibition or loss reduces podocyte motility by inhibiting focal adhesion turnover, thereby preventing proteinuria and effacement [20,21,22]. In response to tubulointerstitial osmotic pressure changes, cells exposed to hypertonic or hypotonic environments activate multiple cellular events to adapt to osmotic disturbances. FAK signaling is activated by osmotic pressure, FAK cascades, ion channel activation, NFAT5-dependent gene expression, and actin dynamics in renal cells [3,14,23,24].

The identification of biomarkers released from renal epithelial cells has the potential to indicate that renal cells are exposed to hyperosmotic stress in the tubulointerstitium and to report abnormal osmotic pressure-induced pathological progression in the kidney. Herein, we evaluated the renal cell secretome exposed to hypertonicity and found that LMO7 (LIM domain protein 7 only) was one of the candidates in the NRK-52E cells secretome exposed to hypertonic medium (620 mOsm/kg). Expression of LMO7, a microfilament-associated protein in the NRK-52E cell secretome induced by hypertonic stimulation in our study, was upregulated in high-salt administered mouse embryonic fibroblasts (MEF) [25,26]. It is a cell–cell junction-associated protein that acts as a transcriptional activity in myogenesis, and its deficiency results in muscle dystrophy [27,28,29,30,31]. It has been identified in podocyte-specific transcripts during disease and in myofibroblast gene expression profiling during fibrosis [32,33]. Physiologically, its cellular functions in the kidney have not been completely elucidated. In our study, we report the cellular function of LMO7 in renal epithelial cells via FAK signaling to sustain the renal epithelial barrier (REB), epithelial integrity, and junctional integrity in renal epithelial cells exposed to hyperosmotic stress.

## 2. Materials and Methods

### 2.1. Antibodies, Chemial Reagents, and Equipment

Antibodies used in this study are: LMO7 (Santa Cruz Biotechnology, Santa Cruz, CA, USA), p-FAK (Cell Signaling, Danvers, MA, USA), Emerin (Santa Cruz Biotechnology, Santa Cruz, CA, USA), β-catenin (Santa Cruz Biotechnology, Santa Cruz, CA, USA), NKCC2 Cell Signaling, Danvers, MA, USA), E-cadherin (Arigo, HsinChu, Taiwan), β-actin (Arigo, HsinChu, Taiwan), ZO-1 (Thermo Fisher Scientific, Waltham, MA, USA), and p-Paxillin (Santa Cruz Biotechnology, Santa Cruz, CA, USA). Specimens were imaged with DM16000B epifluorescence microscopy (Leica, Wetzlar, Germany). Images were analyzed with Software Image-Pro Plus 8.0 (Media Cybernetics, Rockville, MD, USA). Detailed material information is listed in the key resource table (Appendix A).

### 2.2. Animal Care and Experiments

All animals were purchased from the Charles River Breeding Laboratory (Charles River Technology, Bio-LASCO Taiwan Co., Ltd., Taipei, Taiwan). All animals were housed at temperature (22 ± 2 °C), with 50 ± 10% humidity, and an automatically controlled cycle of 12 h light and dark. 

### 2.3. Biopsies Harvest, Specimen Preparation, Histochemical, and Immunohistochemical Staining

Kidney tissues were harvested and soaked in aqueous formaldehyde solution for 24 h. They were dehydrated and embedded in paraffin blocks, which were sliced to a 4 μm thickness using a sliding microtome (SM2125, Leica Biosystems, Nussloch, Germany). The specimens were stained with LMO7 and NKCC2 antibodies to validate their distribution in mouse kidneys. Immunohistochemical staining was conducted according to the manufacturers’ instructions.

### 2.4. Cell and Epithelial Sheet Culture

NRK-52E cells were acquired from the ATCC. These cells were cultured in Dulbecco’s modified Eagle’s medium with 4.5 g glucose/L (DMEM) supplemented with 4% (*w*/*v*) glutamine, 100 u/mL penicillium/streptomycin, and 10% (*v*/*v*) fetal bovine serum, settled as normal DMEM, in an atmosphere of 5% CO_2_/95% air at 37 °C. Serum-free DMEM is a 320 mOsm/kg isotonic medium applied to the epithelial sheet culture. Isotonic DMEM supplemented with 100 mM NaCl and urea was prepared in a 620 mOsm/kg hypertonic medium. The original receipts of hypertonic medium, described in Faust’s protocol [34], were properly modified for NRK-52E cell culture. To form epithelial sheets in a 100 mm Petri dish, 5 × 10^4^ NRK-52E cells were seeded on a 100 mm Petri dish and grown in normal DMEM for four days. NRK-52E cells then formed epithelial sheets on the culture dish. The epithelial characteristics of NRK-52E epithelial sheets were evaluated, with β-catenin and E-cadherin distributed in the border between adjacent cells.

In this study, three osmotic states were applied to assay osmotic responses in NRK-52E epithelial sheets. These were isotonic (320 mOsm/kg), hypertonic (620 mOsm/kg), and hypertonic-isotonic alternation. The hypertonic-isotonic alternation was experimentally conducted in epithelial sheets cultured in 620 mOsm/kg hypertonic medium for an hour followed by 320 mOsm/kg for another hour. Additionally, FAK inhibition was conducted in the presence of 10 µM PF-573228 in the culture medium.

### 2.5. Renal Tubule Cell Primary Culture

The primary renal epithelial cell culture was approved by the IACUC of Kaohsiung Medicine University (document: 108248). Kidneys from C57BL/6 mice were harvested, and the medulla was collected to acquire primary renal epithelial cells. It was minced into small pieces in DMEM (Thermo Fisher Scientific, Waltham, MA, USA). The tissues were then digested in 2 mg/mL type II collagenase in HBSS for an hour, and the cell suspension was sieved using a 70 µm strainer to eliminate undigested pieces. Number of renal cells were counted using a hemacytometer; 2 × 10^4^ cells were seeded on 12 mm coverslip and incubated in DMEM supplemented 5% FBS, 1% penicillin/streptomycin, 1% l-glutamine, 50 mM hydrocortisone, 5 µg/mL insulin, 5 µg/mL transferrin, and 50 nM sodium selenite.

### 2.6. LMO7 Depletion in NRK-52E Cell

LMO7 depletion was conducted using small hairpin RNA (shRNA) transfection. ”ccggACTGCTATCTCCGATTCAAATctcgagATTTGAATCGGAGATAGCAGTtttttg” and ”ccggTAGCAGGTTTGGATAACATAActcgagTTATGTTATCCAAACCTGCTAtttttg” were cloned into the pLKO vector. Two shRNAs, shLMO7#1 and shLMO7#2, targeted to LMO7 transcripts were designed to knockdown endogenous LMO7 in NRK-52E cells. The pLKO.1 Luciferase shRNA was utilized for scrambling. To deplete LMO7 in NRK-52E cells, shRNA mixed with lipofetamine 2000 (Invitrogen, Cergy Pontoise, France) according to manufacturers’ instruction for use. After transfection, cells were harvested and cell lysates were subjected to Western blot analysis to evaluate shLMO7 knockdown efficiency in NRK-52E cells.

In this study, two transfection ways were applied to introduce shRNA to the NRK-52E cells. For transfection of shRNA to adherent cells, 4 µg shRNA and 2 µL lipofetamine were mixed in 2 mL OptiMEM as transfection mixture. Then, the transfection mixture was added to 30% confluent NRK-52E cells in 30 mm Petri dish for transfection for 4 h. After transfection, the transfection mixture was discarded and DMEM medium was added for post-transfection culture. To transfect shRNA to suspended cells, 4 µg shRNA and 2 µL lipofetamine were mixed in 1 mL OptiMEM, and 2 × 10^5^ cells were suspended in 1 mL OptiMEM. The transfection mixture was transferred to 2 × 10^5^ cells, and gently pipette up-and-down to mix cells and transfection mixture in 30 mm Petri-dish. The transfection was last for 4 h. Then, 2 mL DMEM were added for cell culture. Next day, when cells were adhered, medium was discarded and fresh DMEM medium was added to post-transfection culture.

### 2.7. Immunofluorescence Imaging

To validate epithelial integrity, 10^3^ cells were seeded on a 12 mm coverslip and cultured in normal DMEM for four days until epithelial sheet-like formation was observed on the coverslip. Epithelial sheet on 12 mm coverslip were harvested and fixed with 4% paraformaldehyde/0.5% Triton in cytoskeleton buffer (CSK buffer: 10 mM PIPES, 100 mM NaCl, 3 mM MgCl_2_, 1 mM EGTA, and 300 mM sucrose at pH 6.8) for 10 min. Individual antibodies, including LMO7, p-FAK, NKCC2 E-cadherin, and β-catenin, were used to probe the corresponding antigens in cells, and Alexa-546 conjugated phalloidin (Jackson ImmunoResearch Laboratories, West Grove, PA, USA) was utilized to visualize F-actin in cells. Secondary antibodies (Alexa-488 and -546 donkey antibodies; Jackson ImmunoResearch Laboratories, West Grove, PA, USA) against mouse and rabbit IgG were used to display antigens in the cells. A Leica DMi8S epifluorescence microscope (Leica Microsystem, Wetzlar, Germany) equipped with an X-Cite XCT10A (Lumen Dynamics, Wiesbaden, Germany) light source was used to observe and capture fluorescence signals in the cells and kidney specimens.

### 2.8. Immunoblot

Cells were harvested and lysed in RIPA buffer (Merck, Darmstadt, Germany) comprising protease and phosphatase inhibitors (Biotools, New Taipei City, Taiwan). The cell lysate protein concentration was determined using the Bradford protein assay kit (Bio-Rad, Hercules, CA, USA). Then, 30 μg cell lysate proteins were applied on 10 or 12% SDS-polyacrylamide gel for electrophoresis. After electrophoresis, the gel proteins were blotted onto PVDF membranes (Immobilon 0.45m, Merck, Darmstadt, Germany). These membranes were blocked with 5% silk milk for 2 h in TBST buffer (20 mM Tris-Cl, 150 mM NaCl, 0.1% Tween 20, pH 7.4). The membranes were probed with the primary antibody for 2 h. Antibodies against p-FAK, FAK, LMO7, and β-actin were utilized for immunoblotting. The membranes were incubated with the appropriate secondary antibody for another 2 h. Immunoreactive signals were read with an enhanced chemiluminescent reagent (Thermo Fisher Scientific, Pittsburg, PA, USA) using a Bio-Rad ChemiDoc system (BioRad, Hercules, CA, USA). Protein bands were quantified using densitometry software provided by the manufacturer and digitally converted for statistical calculation and analysis.

### 2.9. Quantification in the Fluorescent Imaging

F-actin bundles thickness were measured with Image-Pro-Plus. Merge of LMO7 and cortical F-actin in NRK-52E cells that underwent hypertonic-isotonic alternation for two hours was calculated the yellow color area as numerator and cortical F-actin as denominator. Ratio of LMO-7/F-actin merge to F-actin was calculated. All images were analyzed with software Image Pro Plus 8.0 (Media Cybernetics, Rockville, MD, USA).

### 2.10. Statistical Analysis

The standard error of a statistic was expressed as the standard error of the mean (SEM), and the data were expressed as mean ± SEM. Urinary ACR was expressed as absolute values, and changes were statistically computed using two-way analysis of variance (ANOVA).

## 3. Results

### 3.1. NRK-52E Cells Derived Secretome Profiling

This study aimed to identify proteins released by NRK-52E cells exposed to a hypertonic medium (620 mOsm/kg) (Supplementary method). In the osmotic stress context, 503 proteins were identified in hypertonic secretome profiling, whereas 297 proteins were identified in the isotonic conditioned medium, wherein the NRK-52E cells proliferated for 24 h. In both secretome profiles, 262 proteins were commonly present in hypertonic and isotonic conditioned media (Appendix A). In isotonic conditioned medium, 35 of 297 proteins were uniquely identified in isotonic cell culture (Appendix A). Conversely, 241 proteins were identified in NRK-52E cells cultured in hypertonic medium (Appendix A). The 35 proteins derived from isotonic culture were growth factors, extracellular matrix proteins, cytokines, and extracellular enzymes, but none of the cytoplasmic, nuclear, or plasma proteins were included. Of the 241 proteins from hypertonic conditioned medium, only 47 were counted by Ingenuity pathways analysis (IPA) (Appendix A). Their subcellular localization outside the cell was also evaluated using GeneCards (https://www.genecards.org) with 4 or more confidence score for extracellular release. Most of the 194 molecules of the 241 proteins were predicted, instead of live cells secreted, as proteins from cell debris. Interestingly, 10 of 194 proteins were associated with the plasma membrane (confidence score for plasma membrane association was evaluated using GeneCards with 4 or more) (Table 1). The 10 proteins were majorly categorized as endocytosis-associated proteins, actin dynamics, cell–cell junctions, and plasma membrane flexibility (Table 1). Although 10 were not scored as extracellular proteins, they might be not from cell debris but might be derived from the plasma membrane shedding. LMO7, which is identified in the NRK-52E cells secretome cultured at 620 mOsm/kg (Table 1), was also reported to be a high salt-induced transcript in MEF [25,26]. It seems that LMO7 is an osmotic stress-induced repertoire in MEF and renal cells [25,26]. LMO7, however, is a protein that participates in epithelial morphogenesis, cell–cell junction stabilization, and cytoskeletal assembly [27,29,30,31]. Its cellular function in the kidney in osmotic adaptation or withstanding osmotic pressure was investigated in this study.

### 3.2. LMO Is Present in Renal Tubules in Mouse Kidney

LMO7 expression was reported to be upregulated in podocytes and myofibroblasts in mouse kidneys by gene expression profiling in renal fibrosis and podocytopathies [32,33]. In fact, LMO7 is physiologically identified as a cortical stress fiber-associated protein that localizes with afadin in MDCK cells, MTD-1A, a type of epithelial cell, and embryonic epithelial sheets in *Xenopus* [27,31,35]. Immunohistochemical and immunofluorescent staining with antibody-recognized LMO7 revealed that LMO7 express in a variety of renal cells in mouse kidney (Figure 1A–C). LMO7 expression levels were relatively higher in the region corresponding to a higher osmotic environment, ranging from the cortex and the outer stripe boundary of the outer medulla (OSOM) to the inner medulla papilla tip in the kidney (Figure 1A). This area overlapped with the NKCC2 expression region in the kidney (Appendix A). The LMO7 subcellular localization in renal tubule epithelial cells was dissected using immunofluorescent microscopic imaging. LMO7 was present in the apical portion of renal epithelial cells in renal tubules (Figure 1B). In primary cultured NKCC2 positive renal epithelial cells, LMO7 was predominantly distributed in a cell periphery and at the boundaries between the two junctional cells (Figure 1C). In NRK-5E cells, LMO7 associated with ZO-1 (Zonula Ocludens-1), a tight junction protein (Figure 1D). This was consistent with the LMO7 subcellular localization in MDCK II (Medin-Darby canine kidney (MDCK II) cells [27,35].

### 3.3. LMO7 Associates with Cortical Stress Fibers in Epithelial Sheet Cells

LMO7 association with cortical stress fibers is observed in various epithelial cells [27,28,35]. LMO7 functions in epithelial cells are characterized by a cytoskeleton scaffolding adhesion complex to maintain epithelial integrity and morphogenesis [27,28,36]. To investigate the functional role of LMO7 in the renal epithelial layer, NRK-52E epithelial sheets were established as renal tubular epithelial cell layers (Appendix A). The epithelial characteristics of the NRK-52E epithelial sheet were verified by immunofluorescent staining with antibodies recognizing E-cadherin and β-catenin (Appendix A). In NRK-52E cells, epithelial sheets were cultured in isotonic medium, LMO7 formed thicker fibers around the cell periphery or fine fibers transverse to the cytoplasm (Figure 2A). Following hypertonic stimulation, LMO7 associated with cortical stress fibers in NRK-52E cells cultured in 620 mOsm/kg medium for two hours. When cells were cultured in 620 mOsm/kg hypertonic medium for 24 h, a hypertonicity-damaged epithelial sheet was observed (Appendix A).

The functional role of LMO7 in renal epithelial cells was evaluated with protein depletion by small hair-pin RNA (shRNA) specifically targeted to LMO7 transcript (shLMO7). As per Western blot analysis, the LMO7 expression levels in shLMO7 transfected NRK-52E cells were lower at 48 h after shRNA transfection compared to NRK-52E or NRK-52E cells with scrambled shRNA (Figure 2B,C). shLMO7 transfection can reduce approximately 50% of LMO7 in NRK-52 cell at 96 h post-shRNA transfection (Figure 2C). LMO7 in NRK-52E epithelial sheet cells was associated with cortical actin stress fibers (Figure 2D), which underlies the junctional connection between the two cells. In LMO7 depleted NRK-52E cells, ring-shaped cortical stress fibers were absent and F-actin fibers were disorderly distributed (Figure 2D,E). In NRK-52E cells transfected with scrambled shRNA, cortical stress fibers assembled in the cell cortex and LMO7 was incorporated into them. When LMO7 was depleted in NRK-52E cells, ring-shaped stress fibers assembling around the cell cortex were absent, and short stress fibers were randomly transverse in the cytoplasm (Figure 2E; Appendix A). Epithelial integrity loss was also observed in the LMO7 depleted NRK-52E cells (Figure 2E). LMO7 appeared to participate in cortical stress fiber assembly (Figure 2E,F).

### 3.4. NRK-52E Epithelial Sheet with LMO7 Depletion Is Susceptible to Epithelial Integrity Loss by Osmotic Stress

In epithelial sheets, cortical stress fibers are a type of junctional F-actin assembly and mechanical support to architecture cell shape and organize cell–cell junctions for epithelial integrity [3,28,36,37]. The LMO7 ortholog associates with junctional actomyosin contractile assembly and mediates the junctional connection of two cells to sustain epithelial integrity in *Drosophila* and *Xenopus* [28]. LMO7 expression is inducible by high-salt administration and hypertonic stimulation [25,26]. Herein, the effect of hypertonic stress on LMO7 subcellular localization and its association with cortical stress fibers were evaluated.

NRK-52E epithelial sheets cultured in 620 or 620/320 mOsm/kg osmotic alternation were established as osmotic pressure force pushing plasma membrane inward or pushing plasma membrane inward/outward. The epithelial sheets in osmotic alternation displayed junctional interfaces broadening between two adjacent cells under Nomarski microscopy (Appendix A). This was observed in shLMO7 depleted NRK-52E epithelial sheets cultured in osmotic alternation, wherein junctional integrity loss was apparently visible under Nomarski microscopy (Appendix A). The junctional interface broadening might be a way to investigate individual proteins interplay with the junctional complex associated with contractile actomyosin. It appeared that LMO7 in the epithelial sheet in the isotonic medium was colocalized with fine cortical stress fibers (Figure 3A). When exposed to hypertonic medium for two hours, thicker ring-shaped stress fibers assembled in the cell cortexes in the epithelial sheet and the LMO7 enriched in thicker cortical stress fibers (Figure 3A–D). Hypertonicity can induce F-actin reorganization and LMO7 redistribution in the NRK-52E cells in epithelial sheet. Similarly, when the epithelial sheets were incubated in 620/320 mOsm/kg osmolarity alternating for two hours, the borders between two adjacent cells broadened, although the junctional integrity loss was not apparent (Appendix A). As noted in immunofluorescent imaging, parallel stress fibers assembled around the robust cortical stress fibers in the epithelial sheet interplay and LMO7 associated with the peripheral stress fibers (Figure 3A,E).

Although LMO7 depleted NRK-52E cells did not exhibit severe epithelial integrity loss in epithelial sheets cultured in isotonic or hypertonic media (Figure 3A–D), the LMO7 depleted NRK-52E epithelial sheet in 620/320 osmotic alternated culture for two hours represented a visible epithelial integrity loss (Figure 3E–G). To evaluate the capacity of the LMO7 depleted NRK-52E epithelial sheet to tolerate osmotic pressure force, NRK-52E and LMO7 depleted NRK-52E epithelial sheets were incubated in isotonic, hypertonic, and 620/320 mOsm/kg alternating osmotic medium, respectively. In the LMO7 depleted NRK-52E epithelial sheet, it appeared that two adjacent cells were not tightly adhered together under 620/320 mOsm/kg osmotic alternation (Figure 3E–G). Compromised epithelial integrity in the LMO7 depleted NRK-52E epithelial sheet cultured in hypertonic-isotonic alternation was visible (Figure 3E–G), which implies osmotic pressure force pushes plasma membrane inward/outward and cracked epithelial integrity. The cortical stress fibers were parallel to two disconnecting adjacent cells when LMO7 expression levels were reduced compared to only one cortical stress fiber in the two adjacent cells without LMO7 depletion (Figure 3E; Appendix A).

### 3.5. Hypertonic Induction Results in FAK Activation in NRK-52E Cell Epithelial Sheet

FAK is a non-receptor kinase associated with integrin-based FA complex [38]. Osmotic stress activates FAK in NIH3T3 cells and various kidney cell lines [14,23]. The catalytically active FAK, which are phosphorylated in their tyrosines 576/577, are denoted as p-FAK [39]. To evaluate FAK phosphorylation on tyrosines 576/577 (p-FAK) following hypertonic stimulation in NRK-52E cells, cells were harvested after different hypertonic stimulation durations, with a time increment from 15 min to 12 h. In response to osmotic stimulation, phosphorylated FAK in NRK-52E cells rapidly reached the maximal level within 15 min (Figure 4A,B; Appendix A). The p-FAK levels doubled in cells with 15 min of hypertonic stimulation compared to those without it (Figure 4B). Extending the hypertonic stimulation time, p-FAK levels in the cells gradually decreased. One hour after hypertonic administration, p-FAK levels were lowered to those in cells without hypertonic culture. Hypertonic induction not only represents FAK phosphorylation but is also reflected in FAK localization at FA. As depicted in Figure 4B, the p-FAK based FA was relatively smaller. Additionally, LMO7 expression levels were gradually increased when cells exposed to 620 mOsm/kg with a time increment from 15 min to 3 h (Figure 4C,D; Appendix A). Three hours after, LMO7 expression levels reached to top (Figure 4D). By hypertonic stimulation, as noted in Figure 4C, the bigger p-FAK based FAs were visible in the cells exposed to mOsm 620/kg, compared to p-FAK based FAs in the cells cultured in 320 mOsm/kg (Figure 4E).

### 3.6. FAK Inhibition Results in Fine Cortical Stress Fiber Present in Cell Periphery and Attenuates LMO7 Association with Peripheral Stress Fibers

FAK plays an imperative physiological role in epithelial repair, epithelial permeability and junctional complex assembly regulation, and epithelial integrity maintenance [40,41]. It is also involved in strengthening the endothelial barrier [42]. LMO7 association with robust cortical stress fiber was visualized in NRK-52E epithelial sheet culture in 620 mOsm/kg hypertonic and 620/320 mOsm/kg alternated medium, and FAK was activated via osmotic stimulation. The effect of FAK on LMO7 subcellular localization and cortical actin assembly was evaluated with pharmacological administration of FAK inhibitor PF-573228.

F-actin organization in the NRK-52E cells that were exposed to 620 mOsm/kg with presence of PF-573228 had a fine distorted ring shape, instead of a thick ring-shaped cortical stress fiber in NRK-52E cells cultured in 620 mOsm/kg hypertonic medium without FAK inhibitor (Figure 5A,C). To investigate the effect of FAK inhibition on LMO7 in cells, the LMO7 subcellular localization in cell culture medium with PF-573228 was examined. Less LMO7 was associated with fine cortical stress fibers in NRK-52E cells in the epithelial sheet. The cause of FAK inhibition is not only the fine F-actin assembled in the cell cortex, but also the few LMO7 localized with the fine F-actin in NRK-52E cells exposed to 620 mOsm/kg and 620/320 mOsm/kg alternated osmolarity in the presence of FAK inhibitor (Figure 5A,B).

When the FAK catalytic function was abolished by PF-573228, the NRK-52E epithelial sheet became less sturdy, and FAK inhibition rendered the epithelial sheet fragile. Subsequently, under 620/320 mOsm/kg alternating osmolarity (hypertonic medium 620 mOsm/kg for an hour, followed by isotonic medium 320 mOsm/kg for another hour), epithelial integrity loss was observed in the NRK-52E epithelial sheet in the presence of PF-573228. At 620/320 mOsm/kg osmotic alternation, LMO7 was not able to form fiber-like structures in the boundary between two adjacent cells, and β-catenin was aberrantly distributed with junctional integrity loss (Figure 5B,D). Cells separated from adjacent cells were counted as cells with junctional integrity loss. The number of cells with junctional integrity loss increased in epithelial cell cultures undergoing hypertonic-isotonic alternation in the presence of PF-573228.

### 3.7. LMO7 Depletions Result in Excesses FAK Phosphorylation at Focal Adhesion and Epithelial Integrity Loss in Cells Confronting Osmotic Disturbance

As mentioned above, exposure to hypertonicity or osmotic alteration resulted in FAK activation, cortical actin stress fiber assembly, and LMO7 co-localization with cortical stress fibers. With respect to 620/320 mOsm/kg osmotic alternation, the similar junctional integrity loss phenotype was present in LMO7 depleted and FAK inhibitor-administrated NRK-52E epithelial sheets, which implied that LMO7 and FAK activation participated in maintaining epithelial integrity. In this study, the regulatory relevance of FAK activation and LMO7 functionality was examined in LMO7 depleted NRK-52E cells. With regard to osmotic stress-induced FAK activation in NRK-52E cells with LMO7 knockdown, LMO7 depleted NRK-52E epithelial sheets were exposed to hypertonic-isotonic alternation. p-FAK subcellular localization in LMO7 depleted NRK-52E cells cultured in hypertonic medium were observed in focal adhesions at the cell–cell junction boundaries. Upon osmotic alternation, FAK was phosphorylated and translocated to focal adhesion and disordered robust cortical stress fibers assembled in the cells in the epithelial sheet without LMO7 depletion (Figure 6A). The number of FAK-based focal adhesions was more in LMO7 depleted NRK-52E epithelial sheets exposed to osmotic alteration. Western blot analysis demonstrated that p-FAK levels in LMO7 depleted NRK-52E cultured cells under osmotic alternation were relatively higher than those in NRK-52E cells exposed to osmotic alternation (Figure 6B,C). LMO7 depletion in NRK-52E cells resulted in excessive FAK activation, and junctional integrity loss was present in the LMO7 depleted NRK-52E epithelial sheets (Figure 6A–C).

Since the FAK is a kinase, active FAK can phosphorylate its substrates that are present in focal adhesion (FA) 43. The Paxillin is one of substrate of FAK in FA and is phosphoryalted by active FAK at FAs. To assess enzymatic activity of FAK in FA, the p-Paxillin levels in FAs was investigated. In NRK-52E cells without LMO7 depletion, the p-Paxillin are present in FAs. Consistent with more FAK present in FAs in the LMO7 depleted cells cultured in 620 mOsm/kg medium, more p-Paxillin based FAs formed in boundaries of cell–cells in LMO7 depleted NRK-52E epithelial sheet that were stimulated by 620 mOsm/kg medium (Figure 6D).

### 3.8. Role of FAK in Protecting Renal Epithelial Cell from Hyperosmotic Stress

Volume depletion and cell shrinkage resulted from the cells exposed to hypertonic medium (Appendix A). Conversely, cell swelling occurred in cells cultured in the hypotonic medium (Appendix A). When cells were placed in hypotonic or hypertonic medium for two hours, cell volume changes were observed. p-FAK levels were elevated during the first hour (Figure 3A,B). Upon prolonged administration of hypertonic stress, cell death was observed in the epithelial sheet (Appendix A). The cell death ratio in the epithelial sheets cultured in 320, 420, 520, and 620 mOsm/kg media for 24 h was evaluated by flow cytometry (Figure 7A). Beside FAK regulates actin remodeling and LMO7 redistribution in the cells exposed to hypertonic medium, the enzymatic function of FAK also involves a protective mechanism to save cells from death when cells state in hypertonic environment. Although the enzymatic active p-FAK levels drops down an hour after 620 mOsm/kg hypertonic stimulation, FAK can mitigate percentage of dead cells and FAK inactivation by PF-573228 results in increasing number of dead cells under hypertonic environment. In presence of FAK inhibitor in the 620 mOsm/kg hypertonic medium for 24 hours, the proportion of dead cells elevated to 54%, whereas approximately 40% dead cells were estimated in the cells exposed to 620 mOsm/kg hypertonic stress without presence of PF-573228 by flow cytometric analysis (Figure 7B).

## 4. Discussion

The epithelial barrier is the physiological interface between the external and internal milieu of the body [43,44]. This physiological interface in the kidney comprises a renal tubular epithelial layer [44]. Renal tubular epithelial sheets in the kidney are not only structural isolators but also physiologically regulate body fluid and electrolyte homeostasis [44,45]. Cells in the REB frequently confront to dynamic osmotic change [2]. The osmotic pressure force arises when extracellular change in osmolarity [46,47]. The epithelial cell adaptation to osmotic changes in the kidneys is a fundamental issue. However, the molecular mechanism underlying the adaptation of epithelial cells to osmotic disturbances in the kidneys remains obscure.

LMO7 is associated with adherens junctions (AJs) in renal epithelial cells [27,35,48]. Its functioning in AJs scaffolding the cortical actin cytoskeleton was observed in renal epithelial cells (MDCK cells) [27,35]. In muscle cells, LMO7 has biological significance in myogenesis and muscular function [29,30]. With regard to the LMO7 protein functions in epithelial morphology, Matsuda et al. and Beati et al. have reported that the LMO7 ortholog is involved in epithelial integrity maintenance and morphogenesis in *Xenopus* and *Drosophila* [27,28]. Nevertheless, LMO7 transcripts are increased in cells cultured in high-salt and hypertonic conditions, and its functions in resisting or tolerizing hypertonicity have not been elucidated. LMO7 regulates actomyosin bundle contractile actions in epithelial cells for epithelial integrity [28,31,36]. In fact, the LMO7-associated actomyosin contractile action has mechanical strength [8]. The role of LMO7 in preserving epithelial integrity when the epithelial sheet is exposed to osmotic challenge is yet to be elucidated. In the kidneys, LMO7 was identified in renal tubular cells that live in a milieu with osmotic variation (Figure 1A). The mechanical structure of LMO7, associated with cortical F-actin and the cell–cell junctional complex in maintaining epithelial integrity was explored in this study. To evaluate the junctional organization mechanical strength, we established a hypertonic-to-isotonic alternation-conditioned mechanical effect as an osmotic pressure force pushing the plasma membrane inward/outward. In NRK-52E cells cultured in an isotonic medium, LMO7 had a fiber-like structure and was partly associated with cortical stress fibers (Figure 3A). The hypertonic condition redistributed LMO7 to the cell periphery in a circular manner in NRK-52E cells (Figure 3A). Izumi et al. and Farabaugh et al. reported that LMO7 is upregulated by high-salt induction and hypertonic stimulation [25,26].

LMO7, acting in junctional integrity, was tested in the NRK-52E epithelial sheet cultured in hypertonic medium and hypertonic-to-isotonic alternation (Figure 2A–F; Figure 3A–G). Hypertonicity led to cell shrinkage as the pressure force plasma membrane pushes inward, and hypotonic stress results in cell swelling as the pressure force pushes the plasma membrane outward (Appendix A). Under hypertonic-to-isotonic alternation, osmotic pressure forces push the plasma membrane and its membrane-associated cytoskeleton alternatively inward and outward. Hypertonic-isotonic alternation forces osmotic pressure to push the plasma membrane inward and outward, which implies that physical action is much more powerful than hypertonicity-generated unidirectional pressure force. If the epithelial sheet is not strong enough and cell–cell junctions are fragile, the pressure force pushing the plasma membrane inward/relaxing can tear the epithelial sheet at cell–cell junctions. The LMO7 depleted NRK-52E epithelial sheet appeared fragile and could be cracked by the hypertonic-isotonic alternation generated force, and its junctional integrity was susceptible to pressure forces pushing inward and outward. After LMO7 depletion, fragile epithelial sheets and weaker junctional structures might be attributed to disordered F-actin fibers distributed in junctional boundaries between two adjacent cells, instead of compact F-actin bundles with a lattice-like pattern at the boundaries of cell–cell junctions in the NRK-52E cell epithelial sheet challenged with dynamic osmotic disturbance (Figure 3E–G).

FAK is mechanosensory of extrinsic mechanical forces that cascade FA maturation and actin reassembly [5,39]. Hypertonic and hypertonic-isotonic alternation, cell shrinkage, and cell shrinking-swelling alternation, generate pressure force inward or outward to the actomyosin contractile network in the epithelial sheet cells. By advancing the extrinsic mechanical structure in epithelial cells, osmotic pressure force can activate FAK to drive cytoskeletal remodeling [3,14,24]. When NRK-52E epithelial sheets were exposed to 620 mOsm/kg in the presence of PF-573228, the pressure force-actomyosin contractile-FAK signaling was disrupted, peripheral F-actin exhibited a thin lattice-like pattern, and LMO7 was localized with fine peripheral stress fibers (Figure 5A,C). FAK signaling in actin remodeling increases the mechanical strength of actomyosin/junctional connections from cell to cell. Administration of FAK inhibitor to the NRK-52E epithelial sheet led to fragility and cells became susceptible to osmotic shock-induced cell death under osmotic disturbance. Upon osmotic disturbance, FAK signaling regulates osmotic adaptation for cell survival, cytoskeletal reassembly, and the junctional complex redistribution to hold epithelial cells in the epithelial sheet and to maintain epithelial integrity (Figure 5B,D).

In handling FAK activation by osmotic change, it considers that inappropriate FAK activation compromises epithelial integrity. Inappropriate FAK activation increases cell survival and promotes cell migration and epithelial-mesenchymal transition (EMT) [49,50,51]. EMT and active cell migration are pathogenic factors that cause epithelial integrity loss [52,53]. LMO7 is capable of managing FAK activation and maintaining epithelial integrity. It is constituted in the actomyosin/junctional connections between cells, to hold epithelial cells in the epithelial sheet. When renal epithelial cells encounter hypertonicity, LMO7 is a protein that translocates and associates with peripheral stress fibers. LMO7 depleted NRK-52E cells harbor higher active FAK levels and are susceptible to epithelial integrity loss. Interestingly, LMO7 is not capable of dephosphorylating active FAK. It is possible that LMO7 incorporation into actomyosin/junctional connections might change actomyosin flexibility and elasticity. FAK activation in renal cells, including renal tubular cells and podocytes, frequently drives pathogenic signaling. Active FAK has a non-physiological function in renal epithelial cells for osmotic adaption to conduct physiological functions in the kidney [14]. Indeed, the FAK catalytic activity must be carefully managed since active FAK in epithelial cells pathogenically promotes epithelial-mesenchymal transition (EMT) and cell migration [50,51]. Consequently, epithelial integrity is damaged. As active FAK is enzymatically activated without feedback control, excess active FAK causes EMT or cell mobility in the epithelial sheet. Thereafter, epithelial integrity loss occurs. This signifies that FAK activation in the renal epithelial barrier is a vital factor in renal epithelial integrity (Figure 7C). Although LMO7 is not an enzyme that directly manipulates FAK catalytic activity, it is able to manage p-FAK levels when renal cells meet osmotic stimulation. Our findings shed light on the important role of LMO7 in the adaptation of junctional complexes to actomyosin contractility and revealed that LMO7 coordination with FAK signaling may fail in maintaining microfilament reorganization-modulated junctional integrity in renal epithelia adapted to abrupt hyperosmotic stress.

## 5. Conclusions

When cells confront osmotic change, the osmotic pressure force pushes the plasma membrane inward/outward. In response to osmotic pressure, cells remodel the cytoskeleton organization and build robust mechanical support against the osmotic pressure force. This cellular adaptation can survive under osmotic stress. Inside the kidney, there exist osmotic dynamics of renal epithelial cells. Renal epithelial cells under osmotic disturbance activate FAK signaling to reassemble cortical stress fibers to sustain epithelial morphology and integrity. LMO7 associates with cortical stress fibers and scaffolds by binding to cell–cell junctional complexes. This association has a mechanical structure that forms an epithelial sheet and maintains epithelial integrity. In addition to structural support, LMO7 can manage FAK activation by osmotic pressure force. Our data demonstrated that the mechanical strength of the renal epithelial sheet dynamically adapts to osmotic changes coordinated with FAK signaling, LMO7-associated cortical F-actin assemblage, and cell-junctional complex.

## Figures and Tables

**Figure 1 cells-11-03805-f001:**
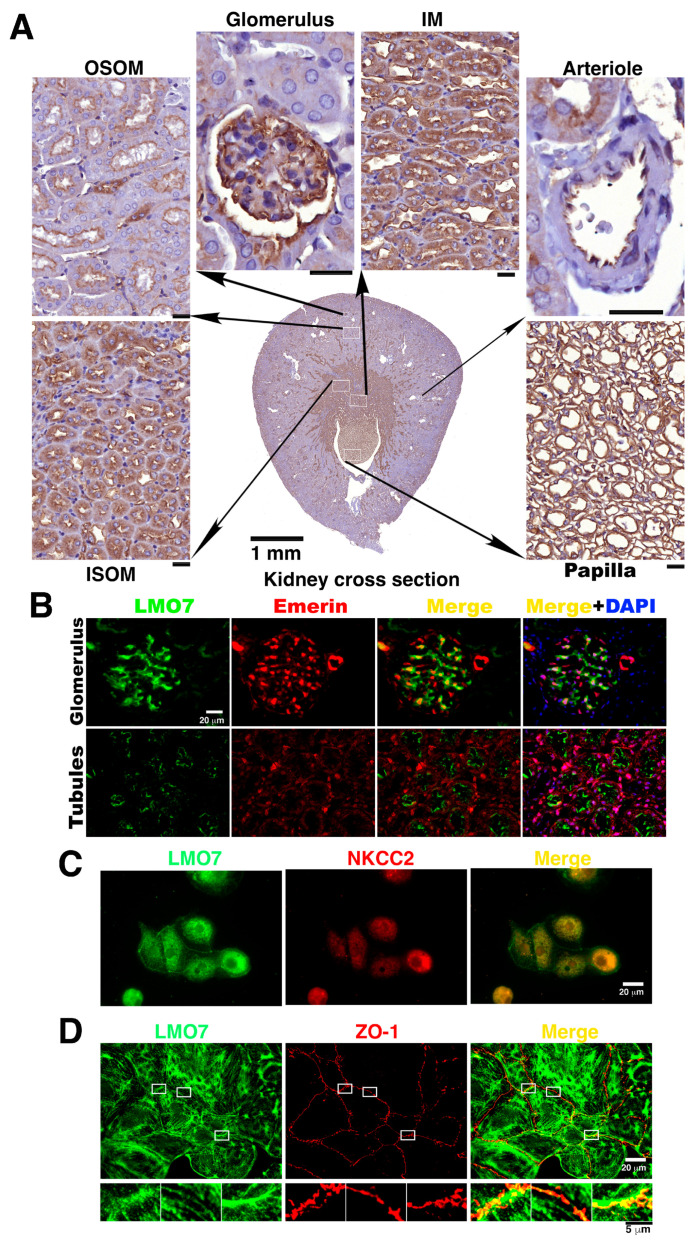
LMO7 expression in kidney. (**A**). LMO7 expression levels are relatively higher in renal cells in the inner stripe of the outer medulla (ISOM), inner medulla (IM), and papilla, and lower in renal cells in the outer stripe of the outer medulla (OSOM). In addition to renal tubules, LMO7 is also detected in the glomerular endothelial cells lining arterioles and podocytes. Bars, 20 µm. (**B**). To visualize the LMO7 subcellular localization in glomeruli and nephritic ducts, kidney specimens are stained with an antibody against LMO7 (green) and an antibody recognizing emerin (red), which is an inner nuclear protein and is characterized as a LMO7 binding partner. LMO7 is detected in the glomerulus and apical portion of the renal tubular cells. (**C**). In NKCC2 positive renal tubular epithelial cells, LMO7 is present in the periphery of cells and boundaries of two junctional cells. (**D**). In NRK-52E cells, LMO7 associates with zonula occludens-1 (ZO-1), a tight junction protein.

**Figure 2 cells-11-03805-f002:**
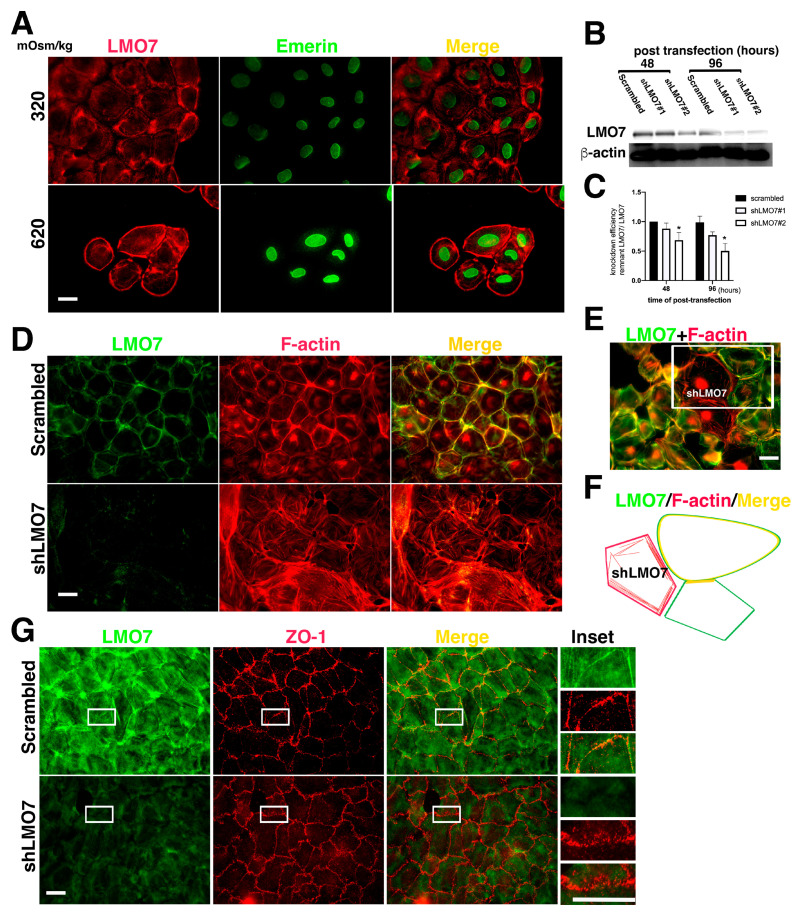
LMO7 associated with peripheral stress fibers in NRK-52E cells. (**A**). LMO7 distributions in NRK-52E epithelial sheets are visualized with antibody-recognized LMO7 (Red) and emerin (Green). In the NRK-52E epithelial sheet, LMO7 forms fiber-like structures and is distributed in the cell peripheries or transversely in the cytoplasm. When cells are exposed to 620 mOsm/kg for 24 h, they are longer in an epithelial sheet, and individual cells isolated from the epithelial sheet are discerned. LMO7 cells cultured in hypertonic medium for 24 h are present in the cell peripheries in a circular style. Bar = 20 µm. (**B**). LMO7 depletion by small hairpin RNA (shRNA) in NRK-52E cells is performed using transient shRNA transfection. Two distinct shRNAs targeting the LMO7 transcript are used to knock down LMO7 expression in NRK-52E cells. shLMO7#1 and shLMO7#2 demonstrate different LMO7 depletion efficiencies. By shRNA transfection, more than half of LMO7 reduces in NRK-52E cells with shLMO7#2, compared to LMO7 expression in NRK-52E cells with scrambled shRNA targeting luciferase. (**C**). Protein band densities are evaluated, and LMO7 protein band density from scrambled is defined as one unit to evaluate other LMO7 band densities. * *p* < 0.05 compared with Scrambled shRNA transfected cells. (**D**). As visualized by F-actin organization in NRK-52E and LMO7 depleted NRK-52E cells, peripheral stress fibers formed in NRK-52E cells are absent in LMO7 depleted cells. In NRK-52E cells, LMO7 associates with cortical stress fibers in epithelial sheet-like NRK-52E cells. (**E**). In NRK-52E cells with different LMO7 expression levels, robust peripheral F-actin cables assemble in cells with higher LMO7 expression and disordered F-actin fibers distributes in the cells with lower LMO7 expression levels. (**F**). A schematic representation of the three cells in the upper right corner of Figure F displays LMO7 merged with robust F-actin cables in cell peripheries and the left cell harbors disordered fine F-actin fibers in the cells harboring few LMO7. (**G**). LOM7 depleted NRK-52E cells and the cells without LOM7 depletion were cultured in 620 mOsm/kg medium for two hours. Zonula occludens-1 (ZO-1) (Red) and LMO7 (Green) in cells were visualized with antibodies, respectively labeled. By osmotic stress, the thicker ZO-1 formed in boundaries of two cells in NRK-52E cells without LMO7 depletion. In LMO7 depleted NRK-52E cells, ZO-1 are aberrantly distributed in boundaries of two cells.

**Figure 3 cells-11-03805-f003:**
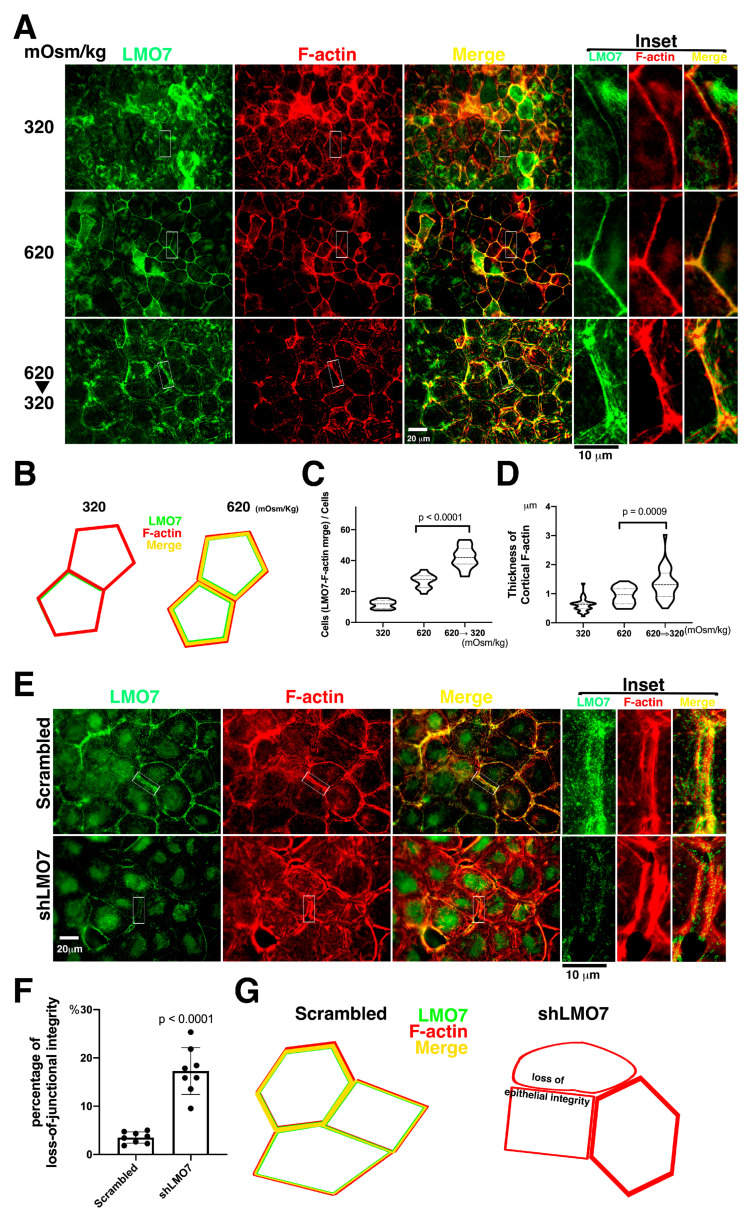
Effect of LMO7 depletion on junctional integrity is susceptible to osmotic pressure challenge. (**A**). The epithelial sheets are cultured in isotonic, hypertonic, and 620/320 mOsm/kg alternating medium. F-actin structures (Red) and LMO7 (Green) distributions are visualized in the epithelial sheet. LMO7 partly associates with fine cortical stress fibers in epithelial sheets cultured in isotonic medium (320 mOsm/kg) for 2 h, and epithelial integrity is preserved. When epithelial sheets are exposed to a hypertonic milieu for 2 h, thickening cortical stress fibers are assembled, and LMO7 proteins associate with robust cortical stress fibers. When the epithelial sheets are confronted with 620/320 mOsm/kg alternated osmolarity (620 mOsm/kg for hour, and 320 mOsm/kg for another hour), cortical stress fibers and their parallel stress fibers assemble in cell peripheries in epithelial-sheet and LMO7 localizes with those stress-fibers. (**B**). Schematic representation demonstrates that thin cortical stress fibers present in the epithelial cell cortex under isotonic 320 mOsm/kg and LMO7 partly associate with cortical F-actin. Exposing to 620mOsm/kg, thicker cortical F-actin assemble in the renal epithelial peripheries, and LMO7 associated with cortical F-actin. (**C**). Hypertonic stimulation promotes LMO7 expression associated with cortical F-actin. The number of cells harboring LMO7/F-actin assembled cortical cytoskeletons is increased in cells grown in hypertonic medium and which undergo hypertonic-isotonic alternation. (**D**). Cortical F-actin fibers assemble in thicker cable when epithelial cells grow at 620 mOsm/kg and undergo 620-to-320 mOsm/kg alternation. (**E**). The epithelial sheets are exposed to osmotic challenge at 620/320 mOsm/kg alternated osmotic milieu and F-actin organization, and LMO7 expression is examined in renal epithelial sheet individual cells. After LMO7 depletion, epithelial integrity is obviously damaged and disordered cortical actin stress fiber organization is visible in the cell peripheries. Additionally, epithelial integrity loss is visible, and a gap between two adjacent cells and junctional connections loss with two parallel cortical stress fibers in cell peripheries are observed. (**F**). The cells that do not tightly adhere together in the visualized view are counted as cells with junctional integrity loss. Most cells in the renal epithelial sheet that undergo hypertonic-to-isotonic alternation are tightly adhered together, and few are not junctional to its adjacent cells in the NRK-52E epithelial sheet. When the LMO7 depleted NRK-52E epithelial sheet is challenged with hypertonic-isotonic alternation, osmotic disturbance causes severe junctional integrity loss in the LMO7 depleted NRK-52E epithelial sheet. (**G**). The schematic diagram displays LMO7 associated with cortical F-actin in cell cortices, and the cells are tightly attached to their adjacent cells. When LMO7 depletes in epithelial cells, they lose their junctional ability to adhere to their adjacent cells.

**Figure 4 cells-11-03805-f004:**
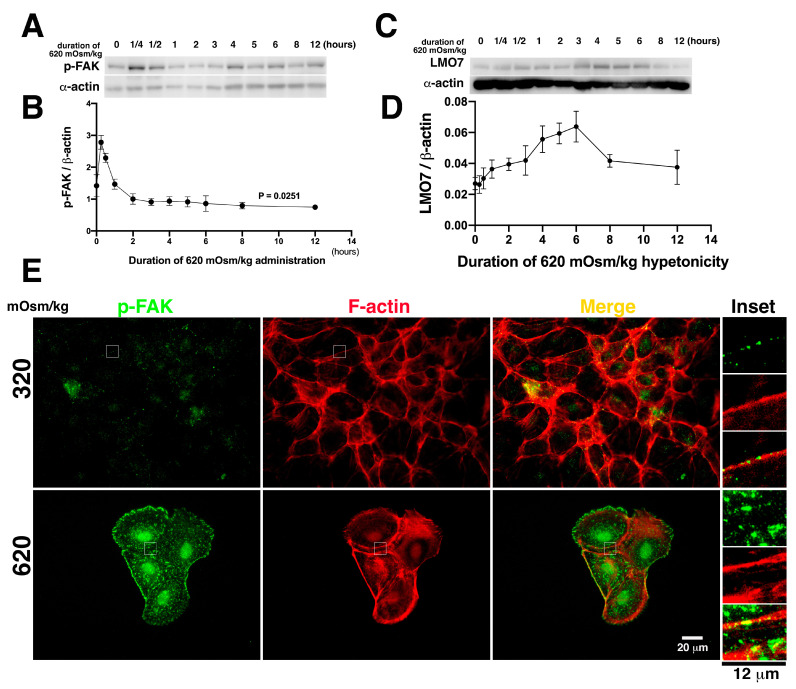
Osmotic induction leads to FAK phosphorylation and its localization at focal adhesion. (**A**). Focal adhesion kinase (FAK) phosphorylated on tyrosine 576/577 (p-FAK) is validated in NRK-52E cells exposed to various 620 mOsm/kg hypertonic medium durations. p-FAK reaches the highest level at 15 min after 620 mOsm/kg exposure. Subsequently, p-FAK levels abruptly decrease. One hour after hypertonic stimulation, p-FAK levels in NRK-52E cells are similar to those in cells without osmotic stimulation. After hypertonic stimulation for 4 h, p-FAK levels remain almost constant in NRK-52E cells. (**B**). The p-FAK levels in NRK-52E cells treated with 620 mOsm/kg hypertonic duration are plotted in a time-dependent curve. The p-FAK levels are highest in cells exposed to 620 mOsm/kg hypertonic medium for 15 min. (**C**). LMO7 expression in NRK-52E cells exposed to 620 mOsm/kg were evaluated by Western blot analysis. (**D**). The LMO7 protein band densities were digitalized and normalized to corresponding β-actin bands. Ratio of LMO7 to β-actin were plotted in a time-dependent curve. (**E**). p-FAK translocates to focal adhesions in NRK-52E cells cultured in 620 mOsm/kg hypertonic medium.

**Figure 5 cells-11-03805-f005:**
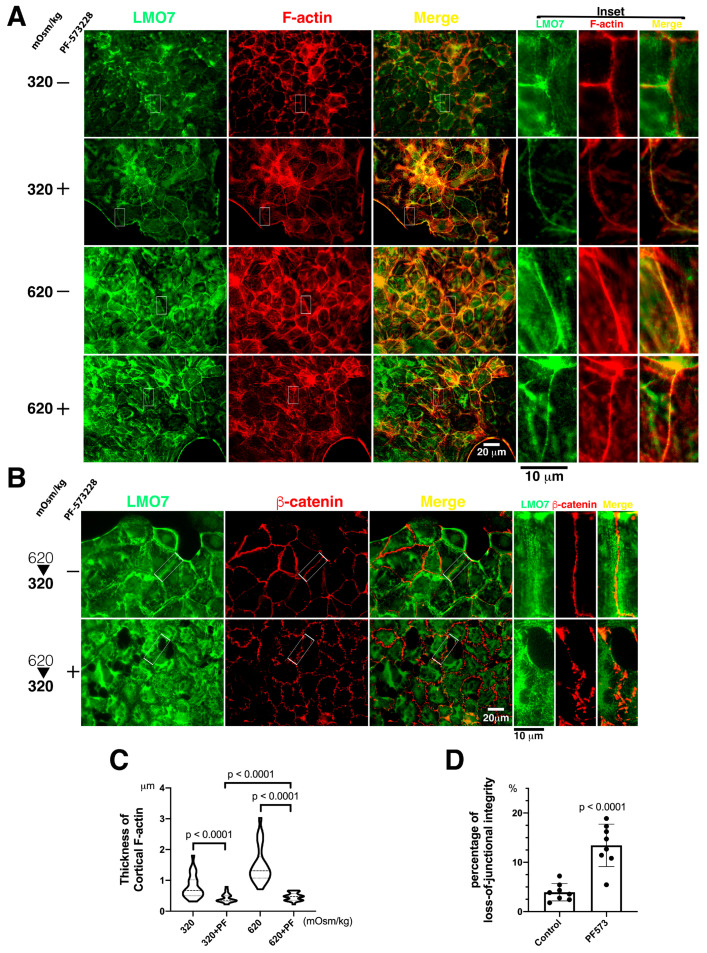
Fine peripheral stress fibers, less LMO7, and junctional integrity loss are present in FAK inhibitor-administrated cells in epithelial sheet. (**A**). The enzymatic function of focal adhesion kinase is blocked by PF-573228. The LMO7 (Green) in NRK-52E cells in epithelial sheet-like cells is fiber-like and is associated with peripheral F-actin (Red) stress fibers or transverses in the cytoplasm without osmotic induction and FAK inhibition. When exposed to hypertonic conditions, cortical F-actin stress fibers assemble in the periphery of cells in an epithelial sheet-like manner, and LMO7 localizes with the cortical F-actin stress fibers. When PF-573228 is present in culture in normal or hypertonic DMEM, the fine cortical F-actin stress fibers in the periphery of cells in the epithelia sheet and LMO7 also represent fine fiber-like structures. FAK inhibition attenuates robust cortical F-actin stress fiber assembly and reduces LMO7-associated cortical F-actin stress fibers in NRK-52E cells. (**B**). The cells are stained with antibodies against LMO7 (Green) and β-catenin (Red). Epithelial sheets settle in an osmotic disturbance altered by the medium osmolarity, which is 620 mOsm/kg hypertonic medium for an hour, followed by 320 mOsm/kg isotonic medium for another hour, which is designated as 620/320 mOsm/kg. Without PF-573228 presence, LMO7 forms a fiber-like structure parallel to the linear β-catenin at the boundary of two cells in the epithelial sheet, confronting osmotic disturbance. When the epithelial sheets are cultured in osmotic disturbance in the presence of PF-573228, parallel LMO7 fibers are absent, and aberrant β-catenin distribution is visible in the boundary between the two cells in the epithelial sheet. Inhibiting FAK enzymatic activity leads to junctional integrity loss in the NRK-52E epithelial sheet in osmotic disturbance. (**C**). FAK activation by osmotic disturbance leads to robust cortical F-actin assembly in cell peripheries and establishes compacted cell–cell junctions in the renal epithelial sheet. The inhibition of FAK enzymatic activity by PF-573228 results in fine cortical F-actin assembly in cells cultured in 320 or 620 mOsm/kg medium. (**D**). Number of cells with lost cell-cell junction and number of total cells in image were counted respectively. Number of cells with lost cell-cell junction divided by total cell number were calculated as percentage of loss-of-junctional integrity. Approximate 2.4% cells do not bear completed junctions in NRK-52E epithelial sheet exposed to 620 mOsm/kg without presence of PF-573228. When FAK inhibitor was added to 620 mOsm/kg hypertonic medium, 12.8% of total cells lost junctional integrity.

**Figure 6 cells-11-03805-f006:**
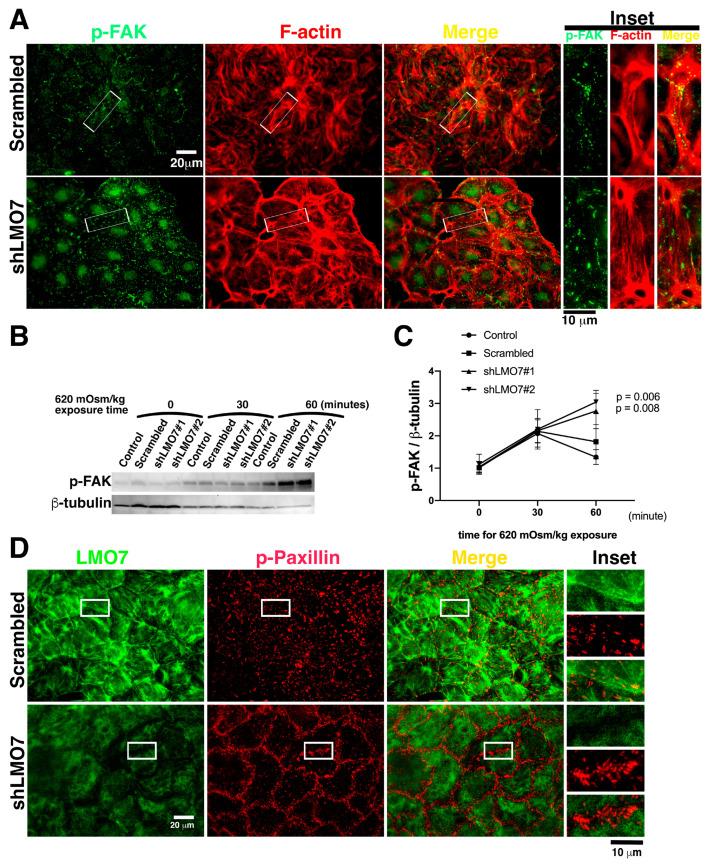
Junctional integrity loss and excessive FAK phosphorylation in LMO7 depleted NRK-52E depleted epithelial sheet, which underwent alternated osmotic challenge. (**A**). The epithelial sheet undergoes hypertonic-isotonic alternation for an hour. Cells in the epithelial sheets are stained with antibodies against p-FAK (Green) and Alexa-568 conjugated phalloidin (Red). In the NRK-52E epithelial sheet without LMO7 depletion, p-FAK is present at focal adhesions (FA), and cortical actin assembles around the cell cortex. The hypertonic-isotonic alternation does not compromise epithelial integrity, and no junctional integrity loss is observed. Conversely, relatively more p-PTK-based FAs assemble and disordered F-actin stress fibers assemble in LMO7 depleted NRK-52E epithelial sheets under hypertonic-isotonic alternation. (**B**). p-FAK levels are higher in LMO7 depleted NRK-52E cells cultured in the medium with 620 mOsm/kg. (**C**). Band densities are digitalized and the ratio of p-FAK to β-actin is calculated. This indicates that p-FAK levels are elevated in LMO7 depleted cells with or without 620 mOsm/kg hypertonic stimulation. (**D**). NRK-52E cells with/ without LMO7 depletion were stained with antibodies, respectively recognized LMO7 (Green) and paxillin phosphorylation on tyrosine 31 designated as p-Paxillin (Red). Without LMO7 depletion, the p-Paxillin based focal adhesions (FAs) are randomly distributed in cells when cells were expose in 620 mOsm/kg. In the LMO7 depleted cells, the 620 mOsm/kg osmotic stimulation resulted in more p-Paxillin present in FAs, and specifically the p-Paxillin based FAs predominately distributed boundary of two cells.

**Figure 7 cells-11-03805-f007:**
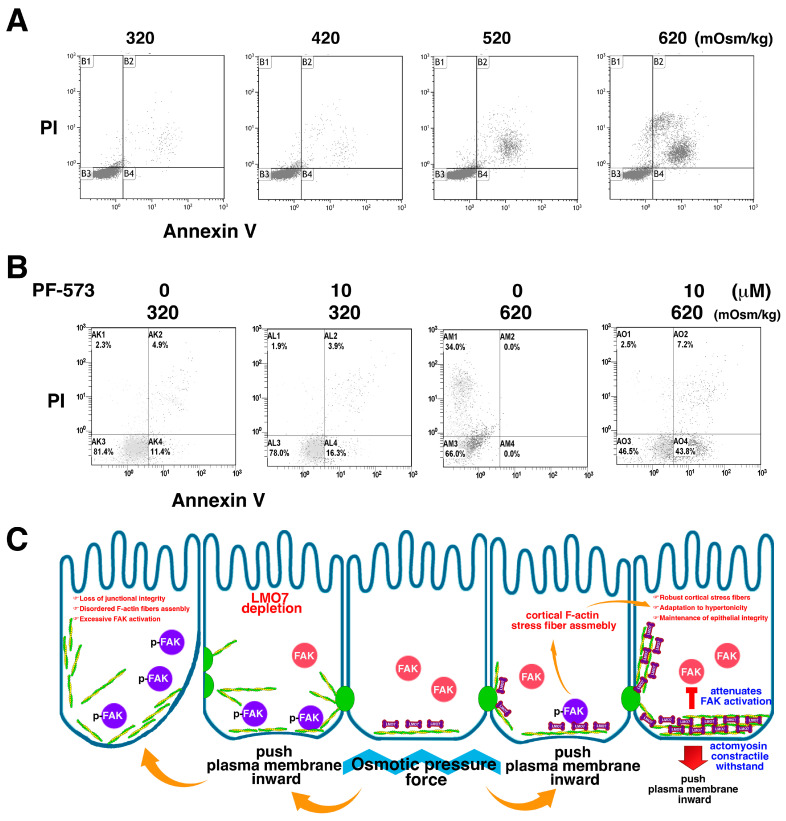
FAK inhibition attenuates osmotic tolerance of NRK-52E cells and osmotic disturbance causes cell death. (**A**). Cells are cultured in 320 mOsm/kg isotonic and hypertonic media at 420, 520, and 620 mOsm/kg for 24 h. In isotonic or 420 mOsm/kg medium, few dead cells are detected using an apoptotic assay with flow cytometry. It appears that 620 mOsm/kg hypertonic stress damages NRK-52E cells and causes cell death. (**B**). FAK signaling regulates cell survival under hypertonic stress. FAK inhibition leads to cell death by hypertonic shock. (**C**). Schematic representation depicts that LMO7 facilitates cortical F-actin stress fiber assembly under hypertonic shock to sustain epithelial integrity. Under hypertonic stress, osmotic pressure pushes the plasma membrane inward and activates FAK. The FAK catalytic activity regulates cortical F-actin reassembly around cell peripheries in the epithelial sheet. LMO7 is an important element incorporated into cortical stress fibers and ordered robust microfilament organization around the cell peripheries. Subsequently, this robust F-actin/LMO7 structure attenuates FAK activation. When LMO7 is absent in epithelial cells, disordered cortical F-actin structure, instead of robust cortical F-actin stress fibers, appear in cell peripheries. The epithelial sheet is fragile when epithelial cells bear disordered cortical F-actin fibers. This disordered F-actin organization in the epithelial cells fails to resist osmotic pressure. As consequence, osmotic pressure force continues to activate FAK, and excessive active FAK promotes pathological epithelial-mesenchymal transition (EMT) and cell migration. Epithelial integrity loss appears in LMO7 depleted epithelial sheet.

**Table 1 cells-11-03805-t001:** Proteins associated with plasma membrane in the NRK-52E secretome exposed to 620 mOsm/kg.

Gene Symbol	Protein Description	Function Description	Score (PM)	Score (CK)
*CTNNA1*	α-Catenin	Associated with cadherin located on plasma membrane	5	5
*CAVIN1*	Caveolae associated protein 1	Caveolae formation and organization	5	2
*CLTA*	Clathrin light chain	Receptor-mediated endocytosis	4	4
*LASP1*	LIM and SH3 protein	Regulation on actin dynamics	3	4
*LIMAI1*	LIM domain and actin binding 1	Actin binding protein involved in actin dynamics	5	5
*LMO7*	LIM domain only 7	Actin binding protein associated with cell–cell junction complex	4	3
*MRC2*	Mannose receptor C type 2	Playing role as endocytotic lectin receptor	4	2
*PICALM*	Phosphatidylinositol binding clathrin assembly protein	Playing role in clathrin-mediated endocytosis	5	2
*SPTA*	⍺-spectrin	Forming cytoskeletal network-associated with membrane	5	5
*STIP1*	Stress induced phosphoprotein 1	Acting as a co-chaperone HSP90AA1	3	2
*TJP1*	ZO-1, Tight-junction protein 1	Linking to tight junction transmembrane proteins	5	4

Confidence scores for proteins associated with the plasma membrane (PM) and cytoskeleton (CK) were evaluated using GeneCards version 5.9. The proteins listed in Table 1 derived from secretome profiling were counted as proteins associated with PM or CK when subcellular localization to PM or its association with CK was scored 4 or more.

## Data Availability

All data used/or analyzed during the current study are available from the corresponding author on reasonable request.

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
