# Peer review of "Coordination of LMO7 with FAK Signaling Sustains Epithelial Integrity in Renal Epithelia Exposed to Osmotic Pressure"

_cells, 2022, doi:10.3390/cells11233805_

Round 1

Reviewer 1 Report

Kidney epithelial cells are in dynamic osmotic environments. Thus, these cells need mechanisms to respond to osmotic pressure forces and remodel cell junctions and cytoskeleton to maintain the epithelial integrity. In this study, Zen et al. identified LIM domain only 7 (Lmo7) as a protein which increased the abundance in response to hypertonic stimulation. They showed that hypertonic stimulation increased the accumulation of Lmo7 and F-actin in cortical stress fibers. Lmo7 depletion resulted in loss of junctional integrity in hypertonic medium. As the underlying mechanism, they showed that FAK activation was required for F-actin and Lmo7 assembly at the cell cortex. Increased FAK activation was observed in Lmo7-depleted cells in a hypertonic environment. They concluded that Lmo7 regulates FAK activation and is responsible for the maintenance of the renal epithelial barrier under osmotic disturbance.

The first half of their results are clean and convincing, while some clarification and quantification are required. The regulatory mechanism between Lmo7/F-actin and FAK is less clear and may require further investigation.

Major comments:

1.     It seems that the majority of the experiments was done in the NRK-52E cells cultured on a regular 100 mm plastic dish, which may not necessarily recapitulate osmotic changes in vivo. I assume that renal epithelial cells are exposed to different osmotic environments between two surfaces of the epithelium; the apical side and the basal side. It might be better to culture cells on a support membrane, such as a transwell, so that the authors can manipulate osmotic conditions only in one side of the epithelial sheet.  

2.     In Fig. 4, the authors describe that FAK phosphorylation was increased within 15 min in response to osmotic stress. How about Lmo7 and F-actin? Is 15 min sufficient to see the increased assembly of Lmo7 and F-actin at the cortex? If Lmo7/F-actin changes require more time, it may suggest that FAK does not directly modulate Lmo7/F-actin. In Fig. 4C, cell densities are quite different between 320 and 620 mOsm/kg. I assume that FAK phosphorylation may change when cells are cultured in sparse or a confluent cell layer.

3.     In Fig. 5, the authors claim that PF-573228 treatment abolishes thick Lmo7/F-actin bundle formation at cell-cell junctions. However, the effects are not easy to see even though quantification is provided. Some arrows or schematics might help. For instance, where is “fine cortical stress fiber”?

4.     In Fig 6, the authors describe excess FAK phosphorylation at focal adhesion in Lmo7-depleted cells under osmotic disturbance. The result might be more convincing if co-IF with focal adhesion markers is provided. p-FAK western blots in Fig. 6B could be improved. Another question is the following: does Lmo7 knockdown increase FAK phosphorylation under normal osmotic conditions?

5.     In the proposed model in Fig. 6F, the authors describe that osmotic pressure force is applied to the basal part of renal epithelium, which modulates basal actomyosin networks. In my understanding, however, the majority of their observation in this manuscript focused on Lmo7 and F-actin at the apical cell-cell junctions, not at the basal side of cells. Please clarify this.

Minor comments:

1.     In Fig. 1B, co-IF of Lmo7 and apical cell-cell junctional markers is missing.

2.     Please clarify what cortical stress fibers are in epithelial cells. Are they actomyosin bundles underlying apical junctions? Or stress fibers connecting focal adhesions at the basal membrane?

3.     In Fig. 2A, cell densities in 320 mOsm/kg and 620 mOsm/kg look quite different, which makes harder to assess the effects of hypertonic stimulation on Lmo7 distribution

4.     The efficacy of Lmo7 depletion by shRNA appears to be weak, especially shLmo7#1 (Fig. 2B,C). Is it because only a subset of cells incorporate shRNA? Or all cells incorporate shRNA but the efficacy of shRNA is very weak.

5.     In Fig. 2D, the authors claim that short stress fibers were randomly transvers in the cytoplasm. Please place arrows to point out “short stress fibers”. In addition, sizes of cells with Lmo7 shRNA look larger than cells with scrambled shRNA. Please comment on that.

6.     Please describe how Fig.2E experiment has been done. Do you co-culture wild-type cells and shLmo7 cells?

7.     In Fig. 2F, please describe differences among these three cells.

8.     In Fig. 3, the authors describe effects of Lmo7 depletion on junctional integrity. However, only Lmo7 and F-actin distribution are shown. Please include IF of other junctional markers, such as ZO-1, b-catenin, E-cadherin.

9.     In Fig. 3C, D and F, please provide the methods of quantification in more detail. For example in Fig. 3C, is it the percentage of cells with the increased assembly of Lmo7 and F-actin at the cell cortex? How did you determine whether or not Lmo7/F-actin assembly was increased in individual cells?

Author Response

Reviewer #1

Comments and Suggestions for Authors:

Kidney epithelial cells are in dynamic osmotic environments. Thus, these cells need mechanisms to respond to osmotic pressure forces and remodel cell junctions and cytoskeleton to maintain the epithelial integrity. In this study, Zen et al. identified LIM domain only 7 (Lmo7) as a protein which increased the abundance in response to hypertonic stimulation. They showed that hypertonic stimulation increased the accumulation of Lmo7 and F-actin in cortical stress fibers. Lmo7 depletion resulted in loss of junctional integrity in hypertonic medium. As the underlying mechanism, they showed that FAK activation was required for F-actin and Lmo7 assembly at the cell cortex. Increased FAK activation was observed in Lmo7-depleted cells in a hypertonic environment. They concluded that Lmo7 regulates FAK activation and is responsible for the maintenance of the renal epithelial barrier under osmotic disturbance.

The first half of their results are clean and convincing, while some clarification and quantification are required. The regulatory mechanism between Lmo7/F-actin and FAK is less clear and may require further investigation.

Major comments:

1.     It seems that the majority of the experiments was done in the NRK-52E cells cultured on a regular 100 mm plastic dish, which may not necessarily recapitulate osmotic changes in vivo. I assume that renal epithelial cells are exposed to different osmotic environments between two surfaces of the epithelium; the apical side and the basal side. It might be better to culture cells on a support membrane, such as a transwell, so that the authors can manipulate osmotic conditions only in one side of the epithelial sheet.  

2.     In Fig. 4, the authors describe that FAK phosphorylation was increased within 15 min in response to osmotic stress. How about Lmo7 and F-actin? Is 15 min sufficient to see the increased assembly of Lmo7 and F-actin at the cortex? If Lmo7/F-actin changes require more time, it may suggest that FAK does not directly modulate Lmo7/F-actin. In Fig. 4C, cell densities are quite different between 320 and 620 mOsm/kg. I assume that FAK phosphorylation may change when cells are cultured in sparse or a confluent cell layer.

3.     In Fig. 5, the authors claim that PF-573228 treatment abolishes thick Lmo7/F-actin bundle formation at cell-cell junctions. However, the effects are not easy to see even though quantification is provided. Some arrows or schematics might help. For instance, where is “fine cortical stress fiber”?

4.     In Fig 6, the authors describe excess FAK phosphorylation at focal adhesion in Lmo7-depleted cells under osmotic disturbance. The result might be more convincing if co-IF with focal adhesion markers is provided. p-FAK western blots in Fig. 6B could be improved. Another question is the following: does Lmo7 knockdown increase FAK phosphorylation under normal osmotic conditions?

5.     In the proposed model in Fig. 6F, the authors describe that osmotic pressure force is applied to the basal part of renal epithelium, which modulates basal actomyosin networks. In my understanding, however, the majority of their observation in this manuscript focused on Lmo7 and F-actin at the apical cell-cell junctions, not at the basal side of cells. Please clarify this.

Minor comments:

1.     In Fig. 1B, co-IF of Lmo7 and apical cell-cell junctional markers is missing.

2.     Please clarify what cortical stress fibers are in epithelial cells. Are they actomyosin bundles underlying apical junctions? Or stress fibers connecting focal adhesions at the basal membrane?

3.     In Fig. 2A, cell densities in 320 mOsm/kg and 620 mOsm/kg look quite different, which makes harder to assess the effects of hypertonic stimulation on Lmo7 distribution

4.     The efficacy of Lmo7 depletion by shRNA appears to be weak, especially shLmo7#1 (Fig. 2B,C). Is it because only a subset of cells incorporate shRNA? Or all cells incorporate shRNA but the efficacy of shRNA is very weak.

5.     In Fig. 2D, the authors claim that short stress fibers were randomly transvers in the cytoplasm. Please place arrows to point out “short stress fibers”. In addition, sizes of cells with Lmo7 shRNA look larger than cells with scrambled shRNA. Please comment on that.

6.     Please describe how Fig.2E experiment has been done. Do you co-culture wild-type cells and shLmo7 cells?

7.     In Fig. 2F, please describe differences among these three cells.

8.     In Fig. 3, the authors describe effects of Lmo7 depletion on junctional integrity. However, only Lmo7 and F-actin distribution are shown. Please include IF of other junctional markers, such as ZO-1, b-catenin, E-cadherin.

9.     In Fig. 3C, D and F, please provide the methods of quantification in more detail. For examplein Fig. 3C, is it the percentage of cells with the increased assembly of Lmo7 and F-actin at the cell cortex? How did you determine whether or not Lmo7/F-actin assembly was increased in individual cells?

Responses:

Major comments:

1. It seems that the majority of the experiments was done in the NRK-52E cells cultured on a regular 100 mm plastic dish, which may not necessarily recapitulate osmotic changes in vivo. I assume that renal epithelial cells are exposed to different osmotic environments between two surfaces of the epithelium; the apical side and the basal side. It might be better to culture cells on a support membrane, such as a transwell, so that the authors can manipulate osmotic conditions only in one side of the epithelial sheet. 

Specific Response:

Thank reviewer for this constructive comment. We have to agree with assumption. It is true that the renal epithelial cell sheet apically exposes to urine, and basal part of the epithelial sheet states in renal interstitial environment. The urinary and renal interstitial osmolarities are not equal. At present, we are going to establish the two osmotic phases culture. This culture system is not perfect for making experimental data, we need to modify.

Two osmotic phases system

The cells were seeded on a semipermeable member. Two different osmotic conditions will respectively be set in central well and up-layer.

2. In Fig. 4, the authors describe that FAK phosphorylation was increased within 15 min in response to osmotic stress. How about Lmo7 and F-actin? Is 15 min sufficient to see the increased assembly of Lmo7 and F-actin at the cortex? If Lmo7/F-actin changes require more time, it may suggest that FAK does not directly modulate Lmo7/F-actin. In Fig. 4C, cell densities are quite different between 320 and 620 mOsm/kg. I assume that FAK phosphorylation may change when cells are cultured in sparse or a confluent cell layer. 

Specific Response:

Thank reviewer for this insightful comment. Expression levels of LMO7 is elevated when cells exposed to 620 mOsm/kg. This Western blot result was arranged in Figure 4 C and D. Thirty minutes after exposure to 620 mOsm/kg, levels of LMO7 is apparently increasing. The LMO7 levels in NRK-52E cells were gradually increased when cells exposed to 620 mOsm/kg. After 8 hours hypertonicity, when LMO7 levels dropped down, cell apoptosis might be the cause. Change in F-actin amount is hard to measure in our lab. Our lab does not have that equipment to monitor F-actin amount in the cells exposed to 620 mOsm/kg. In the original Fig. 4C, when cells exposed to 620 mOsm/kg for 12 hours, hypertonicity caused cell apoptosis and cell density got lower.

3. In Fig. 5, the authors claim that PF-573228 treatment abolishes thick Lmo7/F-actin bundle formation at cell-cell junctions. However, the effects are not easy to see even though quantification is provided. Some arrows or schematics might help. For instance, where is “fine cortical stress fiber”? 

Response:

Thank reviewer for helpful comment. Arrows were arranged to figure 5A and point the thin and severed cortical F-actin in the cells exposed to 620 mOsm/kg with presence of PF-573228. The arrow-heads point thick cortical F-actin in the cells exposed to 620 mOsm/kg without PF-573228.

4. In Fig 6, the authors describe excess FAK phosphorylation at focal adhesion in Lmo7-depleted cells under osmotic disturbance. The result might be more convincing if co-IF with focal adhesion markers is provided. p-FAK western blots in Fig. 6B could be improved. Another question is the following: does Lmo7 knockdown increase FAK phosphorylation under normal osmotic conditions?

Specific Response:

Thank reviewer for this helpful comment. Yes, we have been repeated the osmotic experiments and images will be arranged in Figure 6B.

For the Western blot, we have been repeated experiment, another picture displayed the original Figure 6B, and arranged in figure 6C and D.  

5. In the proposed model in Fig. 6F, the authors describe that osmotic pressure force is applied to the basal part of renal epithelium, which modulates basal actomyosin networks. In my understanding, however, the majority of their observation in this manuscript focused on Lmo7 and F-actin at the apical cell-cell junctions, not at the basal side of cells. Please clarify this.

Specific Response:

Thank reviewer for this comment.

Hypertonicity cause cell volume depletion. The osmotic pressure forces push plasma membrane at apical, lateral, and basal parts inward. When NRK-52E epithelial sheet exposed to 620 mOsm/kg, optically the focal layer at apical plan displayed cortical actin, and LMO7 associated with cortical F-actin. Cells in the renal epithelial sheet is a cubic shape. Our immunofluorescent imagines mainly showed apical surface area. In the area, changes in loss of epithelial integrity, and cortical F-actin reassembly and LMO7 distribution are clearly visualized. Relatively, others focal layers can hardly display such clear changes. For schematic representation, we selectively pictured the transverse section of cuboidal epithelia.  The transverse is contained of molecules distributed in the apical, lateral, basal parts. We can easily explain the molecular relationship.  

Minor comments:

1. In Fig. 1B, co-IF of Lmo7 and apical cell-cell junctional markers is missing.

Specific Response:

Thank reviewer for this comment.

The pictures for co-staining of LMO7 and ZO1 is arranged in Figure 1 D.

2. Please clarify what cortical stress fibers are in epithelial cells. Are they actomyosin bundles underlying apical junctions? Or stress fibers connecting focal adhesions at the basal membrane?

Specific Response:

Thank reviewer for this comment. Yes, the cortical F-actin is an actomyosin bundles. This actomyosin bundles is visualized as closed circle-like pattern by focal layer at the apical surface. The actomyosin bundles in lateral, and basal parts assemble membranous skeletonsliked to adherents junctions, desmosomes in lateral part, and associated with hemidesmosome and focal adhesion complex in basal part. This membrane associated cytoskeleton is as mechanical scaffolds contributed to maintaining epithelial integrity. Mostly, those cytoskeletons do not display and visualize identifiable and regular organizations.

3. In Fig. 2A, cell densities in 320 mOsm/kg and 620 mOsm/kg look quite different, which makes harder to assess the effects of hypertonic stimulation on Lmo7 distribution

Specific Response:

Thank reviewer for this comment. In the Fig. 2A, the pictures were captured at 12 hour the cells exposed to 620 mOsm/kg. Long term hypertonic exposure resulted in cell apoptosis. When cells exposed 620 mOsm/kg for 12 hours, part cells died and cell densities got lower. Experimentally, we initially set the similar cell number in each petri dish. In this study, Figure 2A and Figure 4D are the cells that underwent 620 mOsm/kg hypertonicity for 12 hours. Others figures (Figure 3, 5, and 6), we set the hypertonic exposure for 2 hours.  

4. The efficacy of Lmo7 depletion by shRNA appears to be weak, especially shLmo7#1 (Fig. 2B,C). Is it because only a subset of cells incorporate shRNA? Or all cells incorporate shRNA but the efficacy of shRNA is very weak.

Specific Response

Thank reviewer for this comment. In this study, the shRNA transfection was performed with two different conditions. One is transfection to suspended cells, another transfection was applied to adherent cells. transfection efficiency for suspended cells is much higher than the adherent cells. Beneficially, transfection to adherent cells showed part cells got knockdown and part were not.

The two shLMO7 RNAs that we used to deplete LMO7 in NRK-52E cells did not have the similar knockdown efficiencies. Knockdown efficiency of shLMO7#2 is much better than the shLMO7#1. It appears efficacy of shLMO7#1 is relatively weaker.

5. In Fig. 2D, the authors claim that short stress fibers were randomly transvers in the cytoplasm. Please place arrows to point out “short stress fibers”. In addition, sizes of cells with Lmo7 shRNA look larger than cells with scrambled shRNA. Please comment on that.

Specific Response

Thank reviewer for this comment. Arrows were added to point short F-actin.

Yes, the cells that had LMO7 depletion partly have bigger volume than the cells with scrambled one. We checked other images, and also found similar thing. It could to the LMO7 participates in regulation on cell volumes.

6. Please describe how Fig.2E experiment has been done. Do you co-culture wild-type cells and shLmo7 cells?

Specific Response:

Thank reviewer for this comment. The shRNA knockdown was carried out with transfection ways: the siRNA was applied to suspended cells for 4 hours and the post-transfection cells seeded to petri dish. Efficiency of transfection to suspended cells is much higher but alive cells number was relatively less. The transfection to adherent cells, the transfection shRNA in optiMEM is directly added to adherent cells with 30% conferment for 4 hours transfection. Then, transfection medium was removed and DMEM was added for post-transfection culture for 48 hours. The fig. 2E, transfection was performed to adherent cells. In this study, we did not mixed wild type and LMO7 knockdown cells for cell cultures.

7.  In Fig. 2F, please describe differences among these three cells.

Specific Response:

Thank reviewer for this comment.

8.  In Fig. 3, the authors describe effects of Lmo7 depletion on junctional integrity. However, only Lmo7 and F-actin distribution are shown. Please include IF of other junctional markers, such as ZO-1, b-catenin, E-cadherin.

Specific Response

Thank reviewer for this comment. Yes, we have been repeated the experiments.

9. In Fig. 3C, D and F, please provide the methods of quantification in more detail. For example in Fig. 3C, is it the percentage of cells with the increased assembly of Lmo7 and F-actin at the cell cortex? How did you determine whether or not Lmo7/F-actin assembly was increased in individual cells?

Specific Response:

Thank reviewer for this comment. In this study, we used software “Image ProPlus” to statistically validate merge of LMO7 and F-actin. Merge of LMO7 and F-actin, for example, in NRK-52E that underwent hypertonic-isotonic alternation was calculated the yellow color area as numerator and cortical F-actin as denominator, and ratio of LMO7F-actin-merged to Cortical F-actin was calculated.

Reviewer 2 Report

This is a very complicated paper, but distilled well by the authors.  I don't have any scientific concerns other than the authors should include scale bars on all microscopy images.  There are some figures (e.g., Fig S3 and S5).  

The main concern about the paper is that much of the english language is awkward and not phrased correctly.  While I understood the authors meaning after repeated reading, I believe the manuscript requires some substantial editing to increase clarity.  I suggest sending it to a professional editing service before publication.    

Author Response

F

Round 2

Reviewer 1 Report

The authors have addressed most of my concerns with the original manuscript. I have two suggestions prior to publication. 

1. I still do not quite understand which F-actin staining the authors refer by "a thick ring-shaped cortical stress fiber" or "fine cortical stress fiber" in corresponding figures. Arrows or arrowheads might help.

2. Some of quantification needs additional information. For instance, in Figure 3C and 3D, how many junctions were quantified? Is there a statistically significant difference between 320 and 620?

Author Response

Responses to the reviewer’s comments

We are grateful to the reviewer for the taking the time and effort necessary to review this manuscript again. We sincerely appreciate all valuable comments and suggestion, which helped us to make the quality of the manuscript.

The authors have addressed most of my concerns with the original manuscript. I have two suggestions prior to publication.

  1. I still do not quite understand which F-actin staining the authors refer by "a thick ring-shaped cortical stress fiber" or "fine cortical stress fiber" in corresponding figures. Arrows or arrowheads might help.
  2. Some of quantification needs additional information. For instance, in Figure 3C and 3D, how many junctions were quantified? Is there a statistically significant difference between 320 and 620?

Responses:

Comments:

  1. I still do not quite understand which F-actin staining the authors refer by "a thick ring-shaped cortical stress fiber" or "fine cortical stress fiber" in corresponding figures. Arrows or arrowheads might help.

Response:

Thank you for comment. We have been added arrows to point thicker cortical actin stress fibers in the cells exposed to hypertonic medium and arrowheads to indicate the relatively thin cortical actin stress fibers in the cells exposed to isotonic medium in figure 3A.

  1. Some of quantification needs additional information. For instance, in Figure 3C and 3D, how many junctions were quantified? Is there a statistically significant difference between 320 and 620?

Response:

Thank you for comment. Description on quantification in the immunofluorescent imaging was rewritten and explanations for statistical analysis and calculation were added to materials and methods 2.9 Quantification in the fluorescent images. All p values that are lower than 0.05 were described in corresponding charts.

Reviewer 2 Report

I think the paper is fine to publish now.

Round 3

Reviewer 1 Report

.